# Providing brief information on clinical trials in appropriate formats may improve impressions and willingness to participate among socioeconomically disadvantaged people in France

Salvatore Metanmo[1], Emilien Schultz[1], Michele Planta[1], Sylvain Besle[2,3], Julien Mancini[1,4]*

**1** Aix Marseille Univ, INSERM, IRD, ISSPAM, SESSTIM, Cancer, Biomedicine & Society Group, Ligue Accredited Team, Marseille, France, **2** Centre Léon Bérard, Centre de recherche en cancérologie de Lyon, Institut Convergence PLAsCAN, Université Claude Bernard Lyon 1, INSERM 1052, CNRS 5286, Lyon, France, **3** Human and Social Sciences Department, Centre Léon Bérard, Lyon, France, **4** APHM, Public Health Department (BIOSTIC), Marseille, France

\* julien.mancini@univ-amu.fr

## Abstract

### Background

Clinical trial (CTs) participation rates are low worldwide, including in France. One barrier to participation is a lack of familiarity with CTs, especially in socioeconomically disadvantaged populations. We aimed to ascertain whether providing brief information on CTs to these populations could improve their impression of and participation in CTs.

### Methods

We conducted an online cross-sectional survey among socioeconomically disadvantaged adults living in mainland France. The questionnaire included three parts: 1) routine sociodemographic and socioeconomic questions, health-related variables, and questions on health literacy and numeracy, 2) questions on their impression of CTs and their health, and 3) three questions to assess their comprehension of tables. After completing the questionnaire, participants were randomized into three groups that received an information note on CTs in text (one group) or tabular (two groups) format. The content in tabular format was slightly different for both groups. The same CT questions were asked again to assess possible changes in participants' impression of CTs.

**Data availability statement:** Data Availability: Data cannot be shared publicly because this was not mentioned in the participants consent form. Data available upon request after approval by the Institutional Review Board. The Institutional Review Board is comité d'évaluation éthique de l'Inserm (CEEI, IRB00003888). You can contact the CEEI, Email: ceei@inserm.fr.

**Funding:** The author(s) received no specific funding for this work.

**Competing interests:** The authors have declared that no competing interests exist.

## Results

Among 401 participants, 368 (91.8%) had heard about CTs. Median age of the latter was 40 years, 64% were women, 57% had limited health literacy, and 13% had poor comprehension of tables (score = 0).

After reading the brief CT information note, the median CT impression score increased significantly from 5.0 to 6.5 (p-value < 0.001, effect size r = 0.47) within all sub-groups (e.g., according to the comprehension of tables score, according to health literacy level), except for the sub-group with poor comprehension of tables subsequently given CT information in tabular form.

## Discussion/Conclusion

The initial impression of CTs was quite poor in this socioeconomically disadvantaged sample, but improved after the brief information note on CTs was provided. Information campaigns on CTs could help increase CT participation rates among socioeconomically disadvantaged populations. In this perspective, the information format may be just as important as the content.

## Introduction

Clinical trials (CTs) are research studies in which patients volunteer to help test new ways to treat, diagnose, or prevent diseases. They are used to determine if a new test or treatment works and is safe [1–5]. CTs play a key role in the implementation of evidence-based medicine [2,3]. They are used for numerous reasons, from evaluating treatments to evaluating procedures used to prevent, diagnose or treat diseases, disorders and other health problems [1]. The success of a CT depends on ensuring that enough participants are included, that the sample is as representative as possible of the target population (e.g., breast cancer, people who inject drugs), and that drop-out is minimized [4]. However, there are many barriers to CT enrolment, leading to low participation rates [3,5–7]. One such barrier is low willingness to participate. For example, a study carried out among cancer patients in Florida, USA, reported that only 36.5% were willing to participate in CTs [7], while in France, only 28.0% of persons invited to participate in a COVID-19 vaccination CT were willing [8].

Most studies investigating CT participation rates and willingness to participate have focused on patients in hospital-based samples or samples drawn from registries [5,7,9]. However, the lack of familiarity with medicine is something that grows over time, as studies on health literacy (HL) show [10–12]. For this reason, it is also important to take into account the general public's (i.e., non-patient) impression of CTs, with a view to increasing voluntary participation in general, and in the event of major health crises such as the COVID-19 pandemic. This issue is reflected in a 2020 French study which found that only 47.6% of French adults in the general public were willing to participate in a COVID-19 vaccine CT [13].

A number of studies have investigated the determinants for patients' refusal to take part in CTs. One recurrent determinant is the lack of sufficient knowledge about these trials [5,7,9,14]. In this context, various research teams hypothesized and subsequently demonstrated that providing better information, both in terms of content and format, would increase willingness to participate. For example, a US study showed that providing brief educational material about CTs increased theoretical willingness to take part, particularly among less-educated respondents [15]. Elsewhere, a randomized, online study in the US showed that attitudes towards CTs improved after preparatory training on CTs prior to patients' first oncology visit [14]. A recent systematic review highlighted that one of the reasons explaining why socioeconomically disadvantaged persons are less likely to participate in CTs was their difficulty understanding the purpose of these trials and how they work [16]. This is not surprising given that this population has limited access to health information in general and has poorer HL [17,18].

Giving information is a two-step process. First, trust needs to be built before communicating information [15]. The literature highlights that socioeconomically disadvantaged persons have greater difficulty trusting the healthcare system [19], and that this is one factor why they are less willing to participate in CTs [20–22]. Second, the information to be shared must be in a format that is understandable for the target audience [23–26]. For example, a US study among disadvantaged citizens, found that tabular presentation of numerical data was preferred to icon arrays and was associated with greater comprehension scores [23]. Tabular presentation can also be used to present textual information comparing different options [27]. To our knowledge, no study to date has investigated how well target populations understand content in this format. The present study focused specifically on the second information step listed above.

This study had two objectives: the first was to assess whether providing brief information on CTs improved the general impression of and willingness to participate in CTs in a sample of socioeconomically disadvantaged people in France; the second was to assess the importance of the format of the information shared in terms of ease of understanding.

## Methods

### Type of study

We conducted a cross-sectional online survey of a sub-population of socioeconomically disadvantaged people living in mainland France who were members of a sampling panel (IPSOS i-Say, https://i-say.com/)) between May 14 and May 20, 2018.

### Inclusion criteria:

- Member of the IPSOS i-Say sampling panel,

- Over 18 years of age,

- Resident in mainland France and declaring an annual household income of less than €9000 (single person) or €12000 (households comprising two or more persons). These values were chosen as they reflected potential eligibility for free universal health cover, and corresponded, approximately, to the first decile of income in France [28],

- Volunteering for a 20- to 45-minute online survey.

### Procedures and data collection

After providing consent, participants included in the data analysis were asked to complete a questionnaire divided into three parts:

### First part:

- Routine socio-demographic and socioeconomic data: age, gender, place of residence (city, rural area, town or suburb), level of education, mother tongue, number of people living in household, net monthly household income, recipient or

not of Universal Health Coverage (CSS in French). CSS is social health coverage for persons with lower income (similar to Medicaid in the USA).

- Clinical variables: self-reported health, current chronic condition(s).

- Other health-related variables: confidence in medicine as practiced in France and the single item literacy screener (SILS) to assess functional HL [29,30]. Reflecting the work carried out in a previous study, patients were considered to have limited HL if they answered 'rarely', 'sometimes', 'often' or 'always' to the SILS question [30]. SILS has proven its ability to predict functional autonomy assessed using other self-reported measures [31]. SILS is a valid tool for some rapidly identifying people with limited functional health literacy [32].

- The three-question Subjective Numeracy Scale (SNS3) was used to measure participants' self-assessed ability to use numbers. A validation study has shown that the SNS3 is sufficiently reliable and valid to be used as a measure of subjective numeracy [33]. The SNS3 score is obtained by summing the answers to the three questions, each of which is rated on a scale from 1 to 6 (1 = not at all good or never to 6 = extremely good or very often). In this study, SNS3 is presented as a dichotomous variable: good numeracy (SNS3 score > median) versus poor numeracy (SNS3 score ≤ median).

**Second part:**

- Familiarity with CTs ('Have you ever heard of CTs?'/ 'Have you [or do you know anyone who has] already taken part in a CT?'),

- General impression of CTs and willingness to participate in them. Both these different elements were separately scored from 1 (very negative impression/ resp. not at all willing to participate) to 10 (very positive impression/ resp. very willing to take participate). A score greater than 5 (i.e., the median score prior to the second step of the study when brief information on CVs was provided) was considered to reflect a good impression (resp. strong willingness) to participate.

  All the questions in the second part were taken from a previous survey [15].

**Third part:**

- Comprehension of tables: using the same tool as in a previous US study [23], participants viewed comparative hypothetical post-treatment cancer recurrence risks in three different formats: a table, bar graph, and icon array. For each format, they answered three comprehension questions. For the present study, only the three responses given for the table format were considered. Accordingly, each participant was given a related table comprehension score from 0 to 3. This score is simply the sum of the scores obtained for three questions asked, has not been psychometrically validated and was designed ad-hoc for the a previous US study [23].

  After completing the baseline online questionnaire, participants were provided via the same online platform a brief information note which explained the objectives and procedures of CTs. In order to study whether the form of information could have an impact on the impression and willingness to participate in CTs, we provided information in three forms (corresponding to three groups): In group 1 the information the information given was completely text-based talking about the CTs, and in groups 2a and 2b the information was in the form of a table with 4 questions and answers about CTs. They were then randomly divided into three groups (1, 2a and 2b) (S1 File). The questions and information given differed somewhat between both these groups; group 2b's content focused more on the risks of CTs.

  After reading this information note on CTs, the two initial questions about participants' general impression of and willingness to take part in CTs were asked again, in addition to a question on each of the following elements:

- Difficulty understanding the information note,

- Self-perceived changes in opinion about CTs after reading the information note (nota bene: this question was separate from the objective calculated difference in the pre-post 'CT impression' scores)

- Whether the information provided was new to them or not

- Credibility of the information provided.

## Statistical analysis

Groups 2a and 2b were pooled for statistical analysis. In the remainder of the manuscript, we will therefore refer to two groups: the textual information group (group 1) and the tabular information group (group 2: group 2a and group 2b pooled). Quantitative variables are described using medians and interquartile ranges (IQR), while qualitative variables are described using proportions. The various characteristics of the sample are presented before and after randomization. A p-value (obtained using the chi-squared test for qualitative variables and the Mann-Whitney test for quantitative variables) is associated where appropriate.

Our main outcome was the change in the 'general impression of CTs' score after reading the information note. This change was also evaluated according to two 'comprehension of tabular data' subgroups: those with a comprehension score equal to 0 and those with a score from 1 to 3 (i.e., stratified primary outcome) The different changes were measured by comparing the responses (before and after reading the information material) using a paired Wilcoxon test. The effect size r (Z/sqrt(N)) [34] of the change was also calculated. We also ran linear regression models to study the association between confidence in French medicine and participants' general impression of CTs before and after reading the information note. Analyses were performed using R software version 4.3.1.

## Ethical considerations

The Inserm ethics committee (CEEI, IRB00003888) approved this study. Informed consent was obtained from all participants.

## Results

Among 401 members of the IPSOS sampling panel who completed survey questionnaires, 33 (8.2%) had never heard of CTs and were removed from the analyses. The analyzed study sample therefore comprised 368 individuals.

Median age was 40 years (IQR: 26.8, 53.0) and 42% were aged between 18 and 34 years. Women represented 64% of the sample and 49% lived in a city. In our study, 79% of the participants had an annual net household income ≤9000 €. French was the main language spoken (97%). The majority (61%) had high school or lower education, 57% of participants had limited HL, and 51% had a subjective numeracy score ≤13. Moreover, 11.2% distrusted medicine in France, 47% considered their health to be good, 35% reported having a chronic condition, and 46% received CSS (see above). Thirteen percent scored 0/3 for the 'comprehension of tables' questions. Only 8.4% of the sample had or knew someone who had previously taken part in a CT. Prior to receiving the brief CT information note, the baseline median 'CT general impression' score was 5.0 (IQR; 5.0, 7.0); the baseline median 'hypothetical willingness to participate in a CT' score was also 5.0 (IQR; 3.0, 7.0). Details of these characteristics are shown in Table 1.

### Evolution of impression of CTs and hypothetical willingness to participate in a CT after reading brief information note

Table 2 presents the distribution of variables after receiving the brief information note about CTs. There was no statistically significant difference between the randomized groups. Among the participants, 76% said the information received was

**Table 1. Participants' characteristics according to the randomized format (Textual vs Tabular) of the brief information note on CTs.**

| Characteristic | Textual information, N = 125[1] | Tabular information, N = 243[1] | Overall, N = 368[1] |
|---|---|---|---|
| **Age (years)** | 40.0 (26.0, 53.0) | 40.0 (27.0, 52.0) | 40.0 (26.8, 53.0) |
| **Gender (woman)** | 80 (64%) | 156 (64%) | 236 (64%) |
| **Area of residence** | | | |
| City | 63 (50%) | 117 (48%) | 180 (49%) |
| Rural area | 35 (28%) | 72 (30%) | 107 (29%) |
| Town or suburb | 27 (22%) | 54 (22%) | 81 (22%) |
| **Education level** | | | |
| High school diploma or lower | 84 (67%) | 139 (57%) | 223 (61%) |
| Tertiary diploma | 41 (33%) | 104 (43%) | 145 (39%) |
| **Net annual household income** | | | |
| €0 - €6,000 | 70 (56%) | 117 (48%) | 187 (51%) |
| €6001 - €9000 | 30 (24%) | 72 (30%) | 102 (28%) |
| €9001 - €12000 | 25 (20%) | 54 (22%) | 79 (21%) |
| **French mother tongue** | 122 (98%) | 235 (97%) | 357 (97%) |
| **Needed help reading hospital documents (Functional HL)** | | | |
| Adequate HL | 54 (43%) | 104 (43%) | 158 (43%) |
| Limited HL | 71 (57%) | 139 (57%) | 210 (57%) |
| **Subjective numeracy (median = 13)** | | | |
| High (> median) | 62 (50%) | 118 (49%) | 180 (49%) |
| Low (≤ median) | 63 (50%) | 125 (51%) | 188 (51%) |
| **Trust in French medicine** | | | |
| None/little | 14 (10.8%) | 27 (11.1%) | 41 (11.2%) |
| Some | 42 (34%) | 75 (31%) | 117 (32%) |
| Quite a lot/A lot | 69 (55%) | 141 (58%) | 210 (57%) |
| **General self-perceived health status** | | | |
| Very good | 20 (16%) | 29 (12%) | 49 (13%) |
| Good | 58 (46%) | 115 (47%) | 173 (47%) |
| Fair | 36 (29%) | 75 (31%) | 111 (30%) |
| Poor | 10 (8.0%) | 17 (7.0%) | 27 (7.3%) |
| Very poor | 1 (0.8%) | 4 (1.6%) | 5 (1.4%) |
| Don't know | 0 (0%) | 3 (1.2%) | 3 (0.8%) |
| **Chronic condition** | | | |
| Yes | 40 (32%) | 87 (36%) | 127 (35%) |
| No | 77 (62%) | 134 (55%) | 211 (57%) |
| Don't know | 8 (6.4%) | 22 (9.1%) | 30 (8.2%) |
| **Universal Health Coverage recipient** | 58 (46%) | 110 (45%) | 168 (46%) |
| **Comprehension of tables score (0 to 3)** | | | |
| 0 | 16 (13%) | 31 (13%) | 47 (13%) |
| ≥1 | 109 (87%) | 212 (87%) | 321 (87%) |
| **Have you (or do you know someone who has) taken part in a CT?** | | | |
| Yes | 10 (8.0%) | 21 (8.6%) | 31 (8.4%) |
| No | 110 (88%) | 210 (86%) | 320 (87%) |
| Don't know | 5 (4.0%) | 12 (4.9%) | 17 (4.6%) |
| **Overall impression of CTs (baseline, prior to information, 1–10)** | 5.0 (5.0, 7.0) | 5.0 (5.0, 7.0) | 5.0 (5.0, 7.0) |
| **Positive impression** (score>5) | 59 (47%) | 118 (49%) | 177 (48%) |

*(Continued)*

**Table 1.** (Continued)

| Characteristic | Textual information, N = 125[1] | Tabular information, N = 243[1] | Overall, N = 368[1] |
|---|---|---|---|
| **Willingness to participate in a CT (baseline, prior to brief information on CTs being provided, 1–10)** | 5.0 (3.0, 7.0) | 5.0 (3.5, 7.0) | 5.0 (3.0, 7.0) |
| **Quite likely to participate** (score>5) | 50 (40%) | 114 (47%) | 164 (45%) |

[1] Median (IQR); n (%); HL: Health Literacy.

**Table 2.** Participants' responses after reading the brief information note according to the randomized format of the information provided.

| Characteristic | Textual information, N = 125[1] | Tabular information, N = 243[1] | Overall, N = 368[1] | p-value[2] |
|---|---|---|---|---|
| **Information difficult to understand** | 25 (20%) | 65 (27%) | 90 (24%) | 0.2 |
| **Self-reported change in opinion regarding CTs** (i.e., question asked exclusively after the questionnaire was completed) | | | | 0.2 |
| Much more negative | 4 (3.2%) | 4 (1.6%) | 8 (2.2%) | |
| A little more negative | 5 (4.0%) | 10 (4.1%) | 15 (4.1%) | |
| Unchanged | 72 (58%) | 165 (68%) | 237 (64%) | |
| A little more positive | 38 (30%) | 54 (22%) | 92 (25%) | |
| Much more positive | 6 (4.8%) | 10 (4.1%) | 16 (4.3%) | |
| **Was the information new to them?** | | | | 0.9 |
| Not at all new | 13 (10%) | 45 (19%) | 58 (16%) | |
| Some information was new to them | 78 (62%) | 114 (47%) | 192 (52%) | |
| Most of the information was new to them | 26 (21%) | 66 (27%) | 92 (25%) | |
| Almost all the information was new to them | 8 (6.4%) | 18 (7.4%) | 26 (7.1%) | |
| **Non-credible information** | 16 (13%) | 36 (15%) | 52 (14%) | 0.6 |
| **Overall general impression of CTs after reading information note** (rated from 1 to 10, with 10 indicating a very positive general impression) | 6.0 (5.0, 8.0) | 7.0 (5.0, 8.0) | 6.5 (5.0, 8.0) | 0.5 |
| **Positive general impression of CTs after reading information note** (score>5) | 81 (65%) | 163 (67%) | 244 (66%) | 0.7 |
| **Hypothetically willing to participate in a CT after reading information note (1–10)** | 5.0 (3.0, 8.0) | 6.0 (4.0, 7.0) | 5.0 (4.0, 7.0) | 0.5 |
| **Quite likely to participate in a CT** (score>5) | 56 (45%) | 125 (51%) | 181 (49%) | 0.2 |

[1] n (%); Median (IQR).

[2] Pearson's χ²test, Wilcoxon rank sum test for ordinal variables.

easy to understand, and 86% replied that the information given was credible. A third (29.3%) said their opinion of CTs was positively influenced by reading the information note. More specifically, the percentage of those with a positive general impression of CTs rose from 48% prior to reading the information note (Table 1) to 66% after reading it (Table 2). Similarly, the pre-post percentage of persons willing to participate in a CT rose from 45% (Table 1) to 49% (Table 2).

The median score for participants' general impression of CTs increased moderately overall (from 5 to 6.5 (Fig 1, p-value<0.001, effect size r=0.47). Despite being statistically significant, only a small increase in hypothetical willingness to participate in a CT was observed (median 5 (3, 7) to 5 (4, 7); p-value<0.001, effect size r=0.20; S1 Fig).

We stratified the analysis according to the ability of participants to understand tabular information. Participants with a 0/3 comprehension score for tabular information who subsequently received the information note in tabular format were the only subgroup with no significant increase in the CT impression score (Table 3).

 

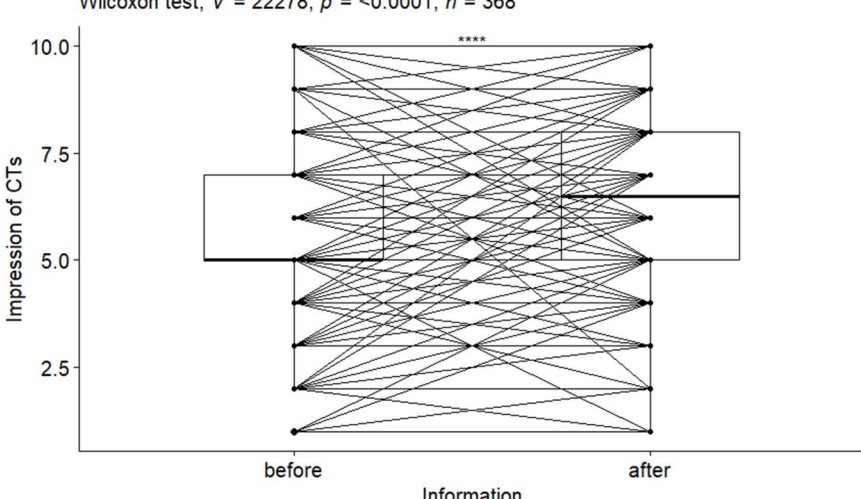

Wilcoxon test, *V* = 22278, *p* = <0.0001, *n* = 368

**Fig 1. Evolution (i.e., pre-post information note) of each patient's general impression of clinical trials.** *Similarly, boxplots are shown to present the quartiles (Q) at each measurement time; we can see that these quartiles (notably $Q_1$, $Q_2$ and $Q_3$) are higher after the information note was read; moreover, the p-value tells us that this difference was statistically significant.*

**Table 3. Evolution of participants' general impression of CTs according to their comprehension of tabular information score and the format of the information subsequently provided after they completed the questionnaire.**

| | Textual information | | | | Tabular information | | | |
| | Median impression of CTs (0–10) | | | | Median impression of CTs (0–10) | | | |
| | Before | After | Effect size | p-value | Before | After | Effect size | p-value |
|---|---|---|---|---|---|---|---|---|
| Comprehension of tabular information | | | | | | | | |
| Low (score = 0) | 5 | 6 | 0.60 | **0.019** | 6 | 6 | 0.09 | 0.75 |
| High (score 1–3) | 5 | 6 | 0.46 | **< 0.001** | 5 | 7 | 0.52 | **< 0.001** |

In order to investigate whether the format-related evolution observed was sensitive to numeracy and HL, pre-post education material changes in participants' general impression of CTs were also measured after stratifying for both these factors. No specific pattern was observed (S1 Table). Trust in French medicine was the only variable significantly positively associated with a favorable impression of CTs irrespective of the study step (i.e., before (β = 0.86 [CI95%, 0.64, 1.1]) or after reading the information note (β = 1.1 [CI95%, 0.94, 1.3])). This was true independent of age, gender, household income, education level, mother tongue, HL level, and randomization group (S2 Table). It was also the only variable significantly associated with a better general impression of CTs after the information note was read (β = 0.27 [CI95%, 0.09, 0.45]).

## Discussion

In our study sample of socioeconomically disadvantaged people living in France, the overall impression of CTs was not good, but improved after participants read the brief information note about CTs. This improvement was associated with the type of information received, and the intrinsic abilities of each individual to cope with different formatting of information.

Fifty-seven percent of our sample had limited functional HL. This large proportion confirms the need for educational interventions to improve health and health-related scientific literacy with a view to limiting health inequalities [17]. The 'general impression of CTs' score in our socioeconomically disadvantaged sample was quite low (median score of 5 before reading information note) compared to that in a study of the French general population (median score = 6.5) using the same scale to measure people's impression of CTs [20]. This difference was to be expected given that the literature has already shown that disadvantaged populations are less knowledgeable about CTs and are therefore less inclined to participate in them [16].

Fewer than half of the participants had a positive score (i.e., > 5) for general impression of CTs and willingness to participate in them (48% and 45%, respectively). In line with findings from a related study conducted in the US [15], participants' general impression of CTs improved significantly after reading the information note. More specifically, 25% of our sample had a more positive opinion of CTs (i.e., calculated on pre-post scores) after reading the information note, while 4.3% had a 'much more positive opinion' (i.e., the self-perceived change in opinion question). Simple information or awareness-raising campaigns to demystify CTs could go a long way towards improving participation rates. In this perspective, repeated campaigns in the healthy general public could be of greater benefit than simply targeting people who are already patients. In our study, although willingness to participate in CTs was still relatively low both before and after reading the information note, a small pre-post increase was seen. It is important to note that there are multiple factors involved in the decision to participate in a CT (presence of a placebo or control group; potential side-effects, physicians' attitudes towards the trial, cost-benefit considerations, etc.) [5,35]. The organization of CTs (e.g., on-site or remote visits) can also impact willingness to participate. For example, a study on an e-cohort of patients with chronic diseases in France regarding their hypothetical participation in a CT found that 90% would participate if the trial were organized according to their preferences, versus 60% if it was not [36].

The only factor significantly positively associated with a favorable general impression of CTs before reading the information note was trust in French medicine. This is consistent with other findings in the literature [21,22], notably Andrews and Davies, who suggested that trust may be the most important factor in the success of research studies, particularly in the context of vulnerable populations [22]. Moreover, after the information note was read, trust in French medicine was associated with an even stronger favorable general impression of CTs. This highlights the need to build trust while providing relevant user-friendly medical information. This is especially important given that only 57% of our sample of socioeconomically disadvantaged population declared they trusted French medicine most of the time.

In our study population, limited functional HL was not associated with participants' impression of CTs, either before or after randomization. This finding might be explained by the fact that the information was brief, and accordingly, respondents might have reported it was easy to understand for this reason (24% reported difficulties, cf. Table 2). Similarly, a previous national study in France reported that cancer patients with limited functional HL were less likely to be invited to participate in CTs, but when they were, their participation rate was no different from those with adequate HL [30]. The problem with this lower invitation rate may lie with doctors who refrain from inviting persons they deem have difficulty understanding CTs to participate, thus creating health inequalities [30]. Indeed, several studies have highlighted that barriers to CT participation are not only attributable to patients, but also to physicians [3,5].

This study highlights the importance of how health information is presented in terms of the target population. More specifically, comprehension of tables was assessed by a score ranging from 0 to 3 obtained by proposing comparative tables for interpretation to study participants. Thirteen percent of participants scored 0 (i.e., did not understand the table at all). Moreover, as expected, participants with poor comprehension of tables better understood the information note on CTs in textual format than the same information given in tabular form.

The improvement in participants' general impression of CTs after reading the information note was observed in all subgroups except – and perhaps unsurprisingly – in persons with poor comprehension of tables who were subsequently given the brief CT information in tabular form. In contrast, as expected, those who had a good comprehension of tables also had a good understanding of the information given in textual format. In other words, it is not enough to provide information

materials to educate disadvantaged populations about CTs; in order to be effective, these materials must take into account users' abilities and understanding, as suggested by other studies [15,23,26,36].

We also looked at changes in participants' general impression of CTs after reading the information note as a function of numeracy and HL level (S1 Table). The fact that the results were similar for all subgroups (i.e., HL, numeracy) supports our hypothesis that it is the format of the brief written information and not a lack of functional HL or of good numeracy which might cause comprehension problems in this disadvantaged population.

This study has limitations arising from the survey method used. First, the study sample may not fully represent the most disadvantaged people in France because of our recruitment process (i.e., using a sampling panel) and potential non-responses of some eligible participants. Second, we had no control group with higher incomes in our study. Studies on the general population are needed in order to compare our results. Finally, we cannot exclude social desirability bias with potential more positive impressions reported after reading the information note.

## Conclusion

Various barriers to CT enrolment are linked to a lack of information or misinformation about these types of trials [7,15,30,37]. This is particularly true among disadvantaged populations. In our study, providing individuals with brief pertinent information helped to improve their general impression of CTs and, to some extent, their willingness to participate in them. Moreover, the way in which this information was provided also had a significant effect on these two outcomes. It is therefore important to adapt information – both in terms of content and format – to the target population.

## Supporting information

**S1 Fig. Evolution of the 'hypothetical willingness to participate in a clinical trial' score after reading the brief information note about clinical trials for the whole sample** . S1 Fig shows the evolution of the median score of willingness to participate in clinical trials after reading the information note on clinical trials (median from 5 (3, 7) to 5 (4, 7); p < 0.001, effect size r = 0.20).
(PDF)

**S1 Table. Evolution of participants' general impression on CTs score according to their health literacy, numeracy, and the format of information on CTs provided.** The evolution of participants' general impression of clinical trials (CTs) according to the type of information received was measured according to participants' health literacy and numeracy level. S1 Table shows the evolution of the general impression of CTs in participants with adequate literacy versus those with limited literacy, and those with high subjective numeracy versus those with low subjective numeracy.
(PDF)

**S2 Table. Linear regression models on participants' general impression of clinical trials (before and after reading brief information note).** S2 Table presents two linear regression models. The variable of interest in the first is the general impression of clinical trials before reading the brief information note, while the second model presents the variables associated with general impression of clinical trials after reading the information note. The third model presents the variables associated with the change in impression after reading the information note.
(PDF)

**S1 File. Information about clinical trials provided to participants.**
(PDF)

## Author contributions

**Conceptualization:** Salvatore Metanmo, Julien Mancini.

**Formal analysis:** Salvatore Metanmo.

**Investigation:** Emilien Schultz, Sylvain Besle, Julien Mancini.

**Methodology:** Salvatore Metanmo, Julien Mancini.

**Validation:** Julien Mancini.

**Writing – original draft:** Salvatore Metanmo, Julien Mancini.

**Writing – review & editing:** Salvatore Metanmo, Emilien Schultz, Michele Planta, Sylvain Besle, Julien Mancini.

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
