## [Decision Letter · Decision Letter 0]

Dear Dr. Mancini,

We look forward to receiving your revised manuscript.

Kind regards,

Vijayaprasad Gopichandran

Academic Editor

PLOS ONE

Journal Requirements:

**Additional Editor Comments:**

Kindly pay close attention to the two reviewers' comments and revise accordingly.

Reviewers' comments:

Reviewer's Responses to Questions

**Comments to the Author**

1. Is the manuscript technically sound, and do the data support the conclusions?

Reviewer #1: Yes

Reviewer #2: Partly

Reviewer #3: Yes

2. Has the statistical analysis been performed appropriately and rigorously?

Reviewer #1: Yes

Reviewer #2: Yes

Reviewer #3: Yes

3. Have the authors made all data underlying the findings in their manuscript fully available?

Reviewer #1: Yes

Reviewer #2: No

Reviewer #3: No

4. Is the manuscript presented in an intelligible fashion and written in standard English?

Reviewer #1: Yes

Reviewer #2: Yes

Reviewer #3: Yes

Reviewer #1: This is a very interesting paper tackling a relevant research area. The methodology used is very appropriate for addressing the intended objectives. The discussions and conclusions were aligned to the results. Ethics approval was sought, and participants had the opportunity to provide consent for participation in line with ethics requirements.

Reviewer #2: Lines 90 & 91, and lines 108-109 – The first objective of the study reads “to assess whether providing brief information on CTs improved the general impression of and willingness to participate in CTs in a sample of socioeconomically disadvantaged” whereas the exclusion criteria say “persons who reported that they had never heard of CTs in the survey questionnaire were 108 secondarily excluded from the analysis.” If people were not provided information, how are they expected to participate irrespective of the method of information? Please explain why this was given as an exclusion factor

Line 100, page 4 – Please provide brief information on what is meant by the IPSOS i-Say sampling panel

Lines 144-149, page 5 – The authors mention 3 groups, but there is no explanation on why there are 3 groups in the study, and what the 3 groups are. Please provide the rationale

Lines 145-147, page 5 – There is no explanation on why a tabular form was chosen for the information regarding CTs. What was the rationale behind choosing the tabular formats?

Lines 145-147, page 5 – Why are there two tabular formats, and why was it pooled later during the analysis?

Lines 160-162, page 6 – Why was group 1 excluded from the analysis? Please provide an explanation

Table 1, page 7 – While the study objective is to see whether there is an improved willingness to participate in CT and the general impression about the CTs, is there a baseline captured on similar information? Without a baseline, how could the comparison be done to say whether such a method of information would improve the participation in CT?

Lines 46-47, page 1 – the Discussion/Conclusion section of the abstract that “The initial impression of CTs was quite poor in this socioeconomically disadvantaged sample, but improved after the brief information note on CTs was provided”. Where is the corresponding data for this? Any baseline assessment percentages and comparison with the post test?

Lines 95-98, page 4 – How did the participant selection happen? Was there any particular method of sampling used to select the participants?

Lines 312-314, page 12 – “In our study, providing individuals with brief pertinent information helped to improve their general impression of CTs and, to some extent, their willingness to participate in them” – does it corresponds to the tabular form or textual form?

Reviewer #3: The study documents the effect of presenting information about clinical trials in different formats (textual vs tabular) on the perceptions and hypothetical willingness of socioeconomically disadvantaged individuals in France to participate in such trials. The research is relevant and timely, addressing an important gap in health communication and equity in clinical trial participation.

Comment 1:

The use of the term “participation” in the title might be misleading, as actual clinical trial participation was not measured. Consider revising the title to reflect the assessment of hypothetical willingness, such as:

“Providing brief information on clinical trials in appropriate formats may improve impressions and willingness to participate among socioeconomically disadvantaged people in France.”

Comment 2:

Please consider adding a brief introduction of the definition of clinical trial for a broader audience.

Comment 3:

On Page 3, line 61, the percentage value “28.0%” is duplicated. Please correct this typographical error.

Comment 4:

The repeated use of the phrase “many studies” in the introduction contributes to a monotonous tone. Rephrase it for improved readability.

Comment 5:

On Page 4, line 89, consider rephrasing the sentence “focused specifically on the second information step listed above” to directly state the study’s objectives.

Comment 6:

Provide a brief description of the IPSOS i-Say panel, including the total number invited, response rate, participant incentives, and how representative the sample is of socioeconomically disadvantaged individuals in France.

Comment 7:

The sentence “Included persons who reported that they had never heard of CTs… were secondarily excluded from the analysis” can be moved from the methodology to the Results section to improve logical flow. The term “secondary exclusion criterion” can be removed.

Comment 8:

Briefly describe the validity of the tools used (e.g., SILS, SNS3, table comprehension score) and their appropriateness for the target population.

Comment 9:

Clarify whether the information notes were delivered via the same online platform as the survey.

Comment 10:

Please confirm whether data were collected anonymously and describe the measures taken to ensure data security and confidentiality.

Comment 11:

On Page 7, line 185, there appears to be a discrepancy regarding the proportion of participants with chronic conditions. Table 1 lists 35%, while the text mentions 37%. Please ensure consistency between tables and text.

Comment 12:

On Page 8, line 200, replace vague fractions such as “three-quarters” with exact proportions or percentages and revise throughout the manuscript (e.g., “two-fifths,” “four-fifths”).

Comment 13:

On Page 10, line 236, please verify and correct any inconsistencies between the regression coefficients mentioned in the text and those reported in Table S2

Comment 14: Consider elaborating more explicitly on potential biases in your study, such as selection bias, social desirability bias, and response bias, particularly given the online, self-reported nature of the survey.

Comment 15:

Reference 10 should be translated and formatted as:

Schultz, É., Ward, J.K., Touzani, R., Rouquette, A. and Mancini, J. (2024). Between Health and Science: Health Literacy and the Perception of Medical Research. Santé Publique, 36(3), 103–108. https://doi.org/10.3917/spub.243.0103

**Do you want your identity to be public for this peer review?** For information about this choice, including consent withdrawal, please see our Privacy Policy

Reviewer #1: No

Reviewer #2: **Yes: ** RAJESWARAN THIAGESAN

Reviewer #3: No

---

## [Author Response · Author response to Decision Letter 1]

24 Jun 2025

Responses to reviewers

To the editor

Dear editor, thank you for your interest in our submission. We have read the reviewers' comments with great interest and have responded point by point to each of them. We have also modified the manuscript wherever necessary to improve the paper for possible publication in your illustrious journal.

Review Comments to the Author

Reviewer #1

This is a very interesting paper tackling a relevant research area. The methodology used is very appropriate for addressing the intended objectives. The discussions and conclusions were aligned to the results. Ethics approval was sought, and participants had the opportunity to provide consent for participation in line with ethics requirements.

Resp : Thank you very much for your comment.

Reviewer #2

Lines 90 & 91, and lines 108-109 – The first objective of the study reads “to assess whether providing brief information on CTs improved the general impression of and willingness to participate in CTs in a sample of socioeconomically disadvantaged” whereas the exclusion criteria say “persons who reported that they had never heard of CTs in the survey questionnaire were 108 secondarily excluded from the analysis.” If people were not provided information, how are they expected to participate irrespective of the method of information? Please explain why this was given as an exclusion factor

Resp : Thank you for your comment, which may help to clarify things. Indeed, the aim of the study was to find out whether providing information about clinical trials (CTs) would improve their impression and willingness to participate. This assumes that those taking part in this analysis have already heard of clinical trials as the question about baseline (cf. infra) impression (and willingness to participate) was only asked to people who already have heard of them. This is why it seemed logical to us to exclude people who have never heard of CTs, as in the pilot study of the questionnaire it was reported that it makes no sense to ask people about their general impression about CTs if they had responded just before that they have never heard of CTs. This is why the questions were filtered and why it was not possible to include those participants in the analysis.

To simplify reading, as suggested by Reviewer #3, we have removed this exclusion criterion from the method section, and left the information relating to the exclusion of these participants in the results section.

Line 100, page 4 – Please provide brief information on what is meant by the IPSOS i-Say sampling panel

Resp : IPSOS i-Say is an online access panel of volunteers recruited by Ipsos (an international research company). Participants in Ipsos iSay surveys earn points and redeem them for gift cards.

Lines 144-149, page 5 – The authors mention 3 groups, but there is no explanation on why there are 3 groups in the study, and what the 3 groups are. Please provide the rationale

Resp : Thank you for your comment. The main hypothesis of our work was that “providing information would improve the impression of CTs” and secondly that the form of this information could have a different impact depending on people's characteristics. The idea behind this is to see whether this information should be adapted to the population studied in order to improve the chance of having good results; this is why 3 groups were randomly constituted to test three different forms of information. In group 1 the information was in the form of a simple text talking about the CTs, and in groups 2 and 3 (renamed 2a and 2b) the information was in the form of a brief comparative table with 4 questions and answers. We initially developed an information table with six questions and answers, but it was not possible to include all six questions at once, as the text would have been significantly longer compared to the text provided to group 1. This is why we formed two subgroups, each with two common and two different questions, including specific information about the potential risks and burden of CT participation in group 2b. We have provided further details in the manuscript as follows and added a translation of the brief information in an Appendix.

"In order to study whether the form of information could have an impact on the impression and willingness to participate in CTs, we provided information in three forms (corresponding to three groups): In group 1 the information given was completely text-based talking about the CTs, and in groups 2a and 2b the information was in the form of a comparative table with 4 questions and answers about CTs."

Lines 145-147, page 5 – There is no explanation on why a tabular form was chosen for the information regarding CTs. What was the rationale behind choosing the tabular formats?

Resp : As mentioned above, the rationale behind this choice lies in the fact that we hypothesized that those who are comfortable with the structure of a table would better understand and be more at ease with this type of information. Asking and answering a question would push them to reason a little, and the fact that it was in a comparative tabular form would enable them to stratify and better organize the information. This format was inspired by options grids developed by Elwyn et al. [1].

Lines 145-147, page 5 – Why are there two tabular formats, and why was it pooled later during the analysis?

Resp : Some people's impression of a subject is often heavily influenced by what they know about the risks associated with it. We made two tabular forms to emphasize the questions about risk that some people in group 2b might have asked themselves. In the end, for this manuscript, we decided to compare only the “textual” form and the “tabular” form, regardless of content; this is why we pooled groups 2a and 2b.

Lines 160-162, page 6 – Why was group 1 excluded from the analysis? Please provide an explanation

Resp : Group 1 was not excluded from the analyses. Groups 2a and 2b were pooled, and in the end the statistical analysis was based on two groups: group 1 and groups 2 (group 2a and 2b pooled. To remove any ambiguity, we have added a sentence to clarify matters.

« In the remainder of the manuscript, we will therefore refer to two groups: the textual information group (group 1) and the tabular information group (group 2: group 2a and group 2b pooled). »

Table 1, page 7 – While the study objective is to see whether there is an improved willingness to participate in CT and the general impression about the CTs, is there a baseline captured on similar information? Without a baseline, how could the comparison be done to say whether such a method of information would improve the participation in CT?

Resp : Thank you for your comment. In Table 1, page 8, we have the baseline data (before providing the brief information on the CTs) with the median for impression and willingness to participate. In Table 2, we have the same information after the provision of the material, and the statistical tests carried out show us that there has been an improvement. Since we're talking about a “feeling”, the reference base for a participant is himself, because we want to know whether his impression of the CTs and his willingness to take part have improved. For willingness to participate, we have been more nuanced, specifying in certain places (Line 188, for example) that it is more a question of “hypothetical willingness” to take part in a CT. We have added the term ‘baseline’ in the text and Table 1 to clarify.

Lines 46-47, page 1 – the Discussion/Conclusion section of the abstract that “The initial impression of CTs was quite poor in this socioeconomically disadvantaged sample, but improved after the brief information note on CTs was provided”. Where is the corresponding data for this? Any baseline assessment percentages and comparison with the post test?

Resp : The corresponding data can be found in Tables 1 and 2, and in Figure 1, which clearly shows a pre-post information evolution.

Lines 95-98, page 4 – How did the participant selection happen? Was there any particular method of sampling used to select the participants?

Resp : As stated in the manuscript, sampling was carried out by IPSOS i-say access panel. The participants already registered in this commercial online panel were selected on a voluntary basis if they matched the inclusion criteria.

Lines 312-314, page 12 – “In our study, providing individuals with brief pertinent information helped to improve their general impression of CTs and, to some extent, their willingness to participate in them” – does it corresponds to the tabular form or textual form?

Resp : As shown in Figures 1 and S1, and specified in the manuscript (lines 214-215), the impression of CTs and willingness to participate improved across the entire population, regardless of the form of information received.

Reviewer #3

The study documents the effect of presenting information about clinical trials in different formats (textual vs tabular) on the perceptions and hypothetical willingness of socioeconomically disadvantaged individuals in France to participate in such trials. The research is relevant and timely, addressing an important gap in health communication and equity in clinical trial participation.

Comment 1: The use of the term “participation” in the title might be misleading, as actual clinical trial participation was not measured. Consider revising the title to reflect the assessment of hypothetical willingness, such as:

“Providing brief information on clinical trials in appropriate formats may improve impressions and willingness to participate among socioeconomically disadvantaged people in France.”

Resp: Thank you for your suggestion. The title of the manuscript has been changed according to your recommendations.

Comment 2: Please consider adding a brief introduction of the definition of clinical trial for a broader audience.

Resp: Thank you for your comment. We have provided a brief definition of clinical trials for the reader.

« Clinical trials Clinical trials (CTs) are research studies in which patients volunteer to help test new ways to treat, diagnose, or prevent diseases. They are used to determine if a new test or treatment works and is [2]. »

Comment 3: On Page 3, line 61, the percentage value “28.0%” is duplicated. Please correct this typographical error.

Resp: Thank you, it's done!

Comment 4:

The repeated use of the phrase “many studies” in the introduction contributes to a monotonous tone. Rephrase it for improved readability.

Resp: Thank you, it's done!

Comment 5: On Page 4, line 89, consider rephrasing the sentence “focused specifically on the second information step listed above” to directly state the study’s objectives.

Resp: Thank you for your comment. We have rewritten according to your recommendations.

Comment 6: Provide a brief description of the IPSOS i-Say panel, including the total number invited, response rate, participant incentives, and how representative the sample is of socioeconomically disadvantaged individuals in France.

Resp: IPSOS i-Say is an online access panel of millions of volunteers worldwide recruited by Ipsos (an international research company). Participants in Ipsos iSay surveys earn points and redeem them for gift cards. The survey was proposed to all members corresponding to the inclusion criteria and was closed as soon as 400 panelists responded. We were not able to ascertain how many panelists received our survey invitation and therefore not able to calculate a response rate. The panel is not perfectly representative of socioeconomically disadvantaged individuals in France as it is based on volunteers able to respond online surveys. However, Ipsos applies rigorous processes (including ongoing monitoring and renewal) to ensure the source of panelists is representative of the general population (including a mix of offline and online recruitment).

Comment 7: The sentence “Included persons who reported that they had never heard of CTs… were secondarily excluded from the analysis” can be moved from the methodology to the Results section to improve logical flow. The term “secondary exclusion criterion” can be removed.

Resp: Thank you for your comment. We have removed this sentence from the method and left it in the results as it was already there.

Comment 8: Briefly describe the validity of the tools used (e.g., SILS, SNS3, table comprehension score) and their appropriateness for the target population.

Resp: Once again, thank you for the relevance of your comment. We have tried to establish the validity of the different scales or scores used in this population, as follows. In addition, we have chosen these short, validated scales were used to maximize the probability of obtaining responses in these disadvantaged populations.

« SILS : SILS has proven its ability to predict functional autonomy assessed using other self-reported measures [3]. SILS is a valid tool for some rapidly identifying people with limited functional health literacy [4].

SNS3 : A validation study has shown that the SNS-3 is sufficiently reliable and valid to be used as a measure of subjective numeracy [5].

Table comprehension score: This score which is simply the sum of the scores obtained for three questions asked, has not been psychometrically validated and was designed ad-hoc for the a previous US stud [6].

Comment 9: Clarify whether the information notes were delivered via the same online platform as the survey.

Resp: After completing the baseline online questionnaire, participants were provided via the same online platform a brief information note which explained the objectives and procedures of CTs

Comment 10: Please confirm whether data were collected anonymously and describe the measures taken to ensure data security and confidentiality.

Resp: The data were collected anonymously. To ensure data security and confidentiality, an anonymous database without IP address or any potentially identifying information was transferred by IPSOS to the research group.

Comment 11: On Page 7, line 185, there appears to be a discrepancy regarding the proportion of participants with chronic conditions. Table 1 lists 35%, while the text mentions 37%. Please ensure consistency between tables and text.

Resp: Thanks for this comment, the data between the text and the table have been harmonized.

Comment 12: On Page 8, line 200, replace vague fractions such as “three-quarters” with exact proportions or percentages and revise throughout the manuscript (e.g., “two-fifths,” “four-fifths”).

Resp: Thank you, it's done!

Comment 13: On Page 10, line 236, please verify and correct any inconsistencies between the regression coefficients mentioned in the text and those reported in Table S2

Resp: Thank you, it's done!

Comment 14: Consider elaborating more explicitly on potential biases in your study, such as selection bias, social desirability bias, and response bias, particularly given the online, self-reported nature of the survey.

Resp: In line with your recommendations, we have added a sentence to the study limits.

« This study has limitations arising from the survey method used. First, the study sample may not fully represent the most disadvantaged people in France because of our recruitment process (i.e., using a sampling panel) and potential non-responses of some eligible participants. Second, we had no control group with higher incomes in our study. Studies on the general population are needed in order to compare our results. Finally, we cannot exclude social desirability bias with potential more positive impressions reported after reading the information note. »

Comment 15: Reference 10 should be translated and formatted as:

Schultz, É., Ward, J.K., Touzani, R., Rouquette, A. and Mancini, J. (2024). Between Health and Science: Health Literacy and the Perception of Medical Research. Santé Publique, 36(3), 103–108. https://doi.org/10.3917/spub.243.0103

Resp: Thank you, it's done!

References

1. Elwyn G, Lloyd A, Joseph-Williams N, Cording E, Thomson R, Durand MA, et al. Option Grids: shared decision making made easier. Patient Educ Couns. févr 2013;90(2):207‑12.

2. Mancini J, Briggs A, Elkin EB, Regan J, Hickey C, Targett C, et al.

---

## [Editor Report · Decision Letter 1]

Providing brief information on clinical trials in appropriate formats may improve impressions and willingness to participate among socioeconomically disadvantaged people in France.

PONE-D-24-54157R1

Dear Dr. Mancini,

We’re pleased to inform you that your manuscript has been judged scientifically suitable for publication and will be formally accepted for publication once it meets all outstanding technical requirements.

Kind regards,

Vijayaprasad Gopichandran

Academic Editor

PLOS ONE

Additional Editor Comments (optional):

Well revised.
---

## [Editor Report · Acceptance letter]

PONE-D-24-54157R1

PLOS ONE

Dear Dr. Mancini,

I'm pleased to inform you that your manuscript has been deemed suitable for publication in PLOS ONE. Congratulations! Your manuscript is now being handed over to our production team.

Kind regards,

on behalf of

Dr. Vijayaprasad Gopichandran

Academic Editor

PLOS ONE